# Dietary Adherence to Recommendations among a Cohort of Adults and Teens with Celiac Disease Maintaining a Gluten-Free Diet Compared to a Nationally Representative Sample: A Cross-Sectional Study

**DOI:** 10.3390/nu16183067

**Published:** 2024-09-11

**Authors:** Jennifer W. Cadenhead, Anne R. Lee, Thanh Thanh T. Nguyen, Benjamin Lebwohl, Peter H. R. Green, Randi L. Wolf

**Affiliations:** 1Program in Nutrition, Department of Health Studies & Applied Educational Psychology, Teachers College, Columbia University, New York, NY 10027, USA; tn2423@tc.columbia.edu (T.T.T.N.); rlw118@tc.columbia.edu (R.L.W.); 2Celiac Disease Center, Columbia University Irving Medical Center, New York, NY 10032, USA; arl2004@cumc.columbia.edu (A.R.L.);

**Keywords:** celiac disease, nutrition, nutrient concerns, nutrient adequacy, excess nutrients

## Abstract

Celiac disease (CeD) is a common autoimmune condition, with a prevalence of ~1%. Currently, a gluten-free diet (GFD) is the only treatment option. Due to fortification rules excluding gluten-free products in the United States of America (U.S.A.), understanding the nutritional adequacy of a GFD is important for promoting optimal health among those with CeD. Cross-sectional examination of multiple 24 h dietary recalls from a study sample of 50 adults and 30 teens with CeD was used to determine nutritional adequacy and excesses according to U.S.A. recommendations. The results were compared with those of 15,777 adults and 2296 teens from a nationally representative sample not reporting CeD, the National Health and Nutrition Examination Survey (NHANES) 2009–2014. Compared with NHANES, our study population was more at risk of low folate and carbohydrate (adults) consumption, and of excessive niacin and vitamin A (teens), as well as saturated and total fat consumption (adults). Overall, though, compared with NHANES, our study participants had similar nutrient concerns but fewer nutritional imbalances, with some notable exceptions. In addition to maintaining a GFD, individuals with CeD should be counseled to maintain a balanced diet and to pay attention to nutrient-dense foods. Special attention should be given to teens in providing dietary counseling to potentially mitigate the risk of future morbidity.

## 1. Introduction

Essential metabolic functions require adequate consumption of certain nutrients and moderation of others. Individuals with celiac disease (CeD), a common autoimmune condition with an estimated prevalence of ~1% [1], are known to be at risk of both inadequate and sometimes excessive consumption of dietary nutrients linked to poor health in the general population [2]. The gluten-free diet (GFD) remains the only available treatment for CeD. Failure to strictly adhere to the GFD may cause individuals with CeD to experience nutrient deficiencies as a result of intestinal damage, which can cause injury to multiple organ systems beyond the gastrointestinal tract [3,4,5,6,7]. Despite known challenges in the GFD [2,8,9,10], it may theoretically be nutrient-dense [11] with appropriate selection of food. However, because gluten-free (GF) products have been perceived as tasting poorly, they are increasingly becoming highly processed to increase their hedonic appeal [12]. As a result, those with CeD have been known to consume high amounts of ultra-processed foods of unknown nutritional value [13]. Ultra-processed items are known to be high in total and saturated fats but low in nutrient density. Importantly, as with other medical diets, those adhering to a GFD are not regularly monitored by the government for nutritional concerns. Thus far, there is little data on potential nutritional imbalances among those with CeD maintaining a GFD in the United States of America (U.S.A.) [14].

Early in the twentieth century, prior to the widespread adoption of the GFD as a treatment for CeD, the U.S.A. incorporated food production standards into federal law. Lawmakers targeted overt population-level nutrient deficiencies resulting from food processing or low dietary intake, aiming to make foods in the market more generally wholesome. These changes were codified in the Code of Federal Regulations, Title 21-FR-21 (Food Code). Specifically, this code details the proportions of ingredients in food formulations as well as the levels of nutrients of concern being fortified or enriched in commonly consumed products. Nutrients like iron, folate, and other B vitamins are among those specified. Many grain-based products, like wheat-flour breads and other gluten-containing foods, were (and still are) among the most commonly consumed staple products and, thus, were the primary products targeted for fortification. In addition, because the concern was for the wellbeing of the general population, specialized products designed for medical diets, like the GFD for individuals with CeD, were not included and, even now, are not subject to these requirements. Therefore, many GF products in the U.S.A. are not fortified.

As an added concern, some GF products—especially those made with processed grains, like refined white rice—are of lower nutritional quality and could be made with healthier choices [2,11,15]. In addition, because GF products have to compensate for the structure and mouthfeel of gluten, GF products may include excessive sodium, gums, and enzymes that adversely impact their nutritional profiles [16], emulsifiers (mono- and diglycerides of fatty acids) or other types of oils/fatty acids, or other ingredients that increase their total and saturated fat content, with concerns for adverse changes in the gut microbiome [17]. Review articles have also shown that individuals with CeD adhering to a GFD have low or inadequate consumption of multiple nutrients, including protein, dietary fiber, vitamin A, folate and multiple B vitamins (especially B12), vitamin C, vitamin D, calcium, iron, magnesium, selenium, and zinc. There have also been concerns about excessive consumption of energy, total fat, saturated fat, added sugars, iron (children), and calcium (children) [2,15,18,19,20,21,22].

There is a need for updated research on the diets of those with CeD who maintain a GFD. This group may not have their nutrient deficiencies recognized, especially as compared with the general American population, who benefit from the Food Code. The last report on how well individuals with CeD on a GFD conformed to U.S.A. nutrient recommendations was published nearly 20 years ago in 2005 [23], and the reporting limited the nutrients examined to fiber, iron, and calcium. This 2005 report also did not compare nutrient adherence to that of the general population. Since then, many products, with varying levels of processing and fortification, have entered the market, and nutrient analysis software has improved.

This research is also important for those with CeD. This research would provide insight into the metabolic risks associated with the GFD, potentially requiring updates to the patient recommendations for individuals with CeD. A recent study looked at population-wide nutrient intake for individuals with CeD in the National Health and Nutrition Examination Survey (NHANES) [24]. Although that study reported the amount of nutrients included in the diet among those with CeD, it did not address adherence to U.S.A. dietary recommendations. It could be helpful to have an understanding of the frequency of nutrient deficits or excesses in order to counsel patients.

Additionally, a limitation of using NHANES data for studying individuals with CeD is that many GF products are not included in many databases used to determine the nutrient profile. GF products’ nutrient profiles in standard nutrient analysis software are often based on using the “typical” food source for a product [25]. For instance, GF pasta may be assumed to be derived from white rice. Newer products may have a broader selection of ingredients, like GF pastas made with brown rice or lentils, which are more nutrient-dense than those reflected in the database. Therefore, we closely examined individual products, based on ingredients, to replicate nutrient label profiles among our sample of individuals with CeD adhering to a GFD.

This study compares a sample of individuals with CeD who had adhered to a GFD for at least a year to a weighted, nationally representative sample, so as to understand their compliance with U.S.A. nutrient recommendations, as provided through the latest national guidance. This will assist in determining whether there may be dietary concerns for individuals with CeD who are adhering to a GFD living in the U.S.A., where the federal policy for addressing population-wide nutrient deficiencies is through selective fortification of mostly wheat-based products. The results of this study may be useful to practitioners advising patients while navigating a GFD. Therefore, the primary aim of this study was to understand the nutritional profiles of individuals with CeD maintaining a GFD, identify any nutritional deficits or excesses based on national recommendations, and provide informed patient recommendations.

## 2. Materials and Methods

### 2.1. Methods and Recruitment

We determined the nutrient requirements of a cross-sectional sample of adults and teens with CeD. Prior to starting, we obtained approval through the Institutional Review Boards at the Columbia University Medical Center and Teachers College, Columbia University. To determine whether our sample might face nutrient concerns by maintaining a GFD, we compared their data to those of participants from NHANES. The National Center for Health Statistics Research Ethics Review Board approved NHANES. Columbia University considers the study of NHANES data to be exempt from IRB review.

We enrolled participants for our study (“Study”) through the Celiac Disease Center of Columbia University Medical Center in New York City (“the Center”). We recruited our study participants primarily through emailing a recruitment message with two follow-up messages to those affiliated with the center via its network of ~5000 members between March and August 2016, including patients and interested family members. Our eligibility criteria included (1) self-reported, biopsy-confirmed CeD for ≥1 year, and (2) the willingness and ability to participate in three cumulative interviews over four to six weeks. Our exclusion criteria included (1) individuals who reported a diagnosis of CeD <1 year, (2) individuals who reported self-diagnosed CeD or whose diagnosis of CeD was based on serological evidence only, and (3) individuals younger than 13 years. We planned to target 30 study adults (age 18+ years) and 30 study teens (age 13–17 years). We enrolled 30 eligible study teens (45 responded, 14 ineligible: 11 lacked duodenal biopsy, 3 lost to follow-up, 1 other reasons). Our adult enrollment exceeded expectations. We enrolled 50 study adults (78 responders; 28 ineligible: 6 no duodenal biopsy, 20 lost to follow-up, 2 other reasons). We provided the study teens with gift cards worth USD 25 as an honorarium. Prior to data collection, we obtained written informed consent or assent; see Appendix A, Figure A1.

NHANES uses sophisticated sampling techniques, incorporating census data and oversampling of specific groups, to obtain nationally representative results of the health and nutritional status among non-institutionalized U.S.A. residents. To calculate statistics using NHANES data, the analysis must apply a survey weighting factor to each participant’s data that takes into consideration information such as their individual probability of being chosen based on gender, race/ethnicity, age, socio-economic status, urbanicity and region of the country, over-sampling to ensure adequate representation of targeted groups in specific census tract strata, and in what stage of the survey they participated. Protocols for NHANES enrollment have been previously described [26]. NHANES sampling occasionally asks questions to assess the health of individuals with CeD among the general population.

With protocols for dietary management of NHANES data [26,27,28,29,30,31,32,33,34,35,36,37,38,39,40,41,42,43,44,45,46,47,48], we used three 2-year cycles, where each 2-year cycle contained ~10,000 participants (2009–2010, 2011–2012, and 2013–2014, *n* = 29,404, representing nearly 300 million). These years were selected because they were the latest and nearest to when our study sample’s data were collected that also included an indication for having been diagnosed with CeD. The primary comparison group was among NHANES participants who did not indicate that they had been diagnosed with CeD, nor had any serology suggestive of CeD, regardless of adherence to a GFD (NHANES adults *n* = 15,777, NHANES teens *n* = 2296). To determine estimated nutrient needs, we relied upon recorded age, gender, weight, and height, further reducing the NHANES adult population *(n* = 15,616). In total, this represented *n*~225 million adults and *n*~20 million teens. We excluded participants who were younger than 13 years (*n* = 7688), pregnant or being breastfed (*n* = 179), who did not have at least one reliable 24 h recall (*n* = 2861), or who had consumed aberrant energy amounts (±2.5 std dev for gender mean) (*n* = 519) (Appendix A, Figure A2).

To assess generalizability, our study sample compared NHANES participants who indicated having a prior CeD diagnosis and affirmed maintaining a GFD in the medical questionnaire. We excluded NHANES participants with serological indicators suggestive of undiagnosed or newly diagnosed CeD, regardless of whether they were adhering to a GFD. Individuals were considered to potentially have had active and/or non-adherent, recently diagnosed, or undiagnosed CeD if positive for both IgA-tTG ≥ 4.0 U/mL and IgA EMA (“CeD and GFD”, NHANES adults *n* = 25, representing *n*~533 K adults). We excluded an analysis of NHANES teens with prior CeD due to the small actual number identified (*n* = 1, representing *n*~19 K).

This categorization was similar to prior researchers’ methods in categorizing CeD status among NHANES participants [49].

### 2.2. Demographics and Medical History

We collected the following self-reported demographic data on our study sample: age, self-reported gender, education (highest attained), and home address (zip code). We also collected self-reported medical variables from our study sample, including height, weight, time since CeD diagnosis, and history of visiting registered dietitian nutritionists.

NHANES demographic variables included gender, age, education, and percentage of income to poverty level. NHANES medical variables included prior CeD diagnosis, GFD status, CeD serology (IgA-tTG and IgA EMA), height, and weight.

### 2.3. Diet Assessment

#### 2.3.1. Twenty-Four Hour (24 h) Recalls

We collected three 24 h dietary recalls (24 h recalls) from our study sample. We obtained the first recall at the initial interview. During the first interview, prior to the recall, we provided our study sample a folder with paper diagrams depicting commonly consumed foods and drinks, examples of typical serving sizes, and a ruler. We encouraged them to take the items home for use during the follow-up recalls. To capture at least one weekend and one weekday recall, we completed the remaining recalls via unannounced phone calls, where all were completed ~2–3 weeks apart. We followed the validated multiple-pass method [50,51]. Trained interviewers, including J.C. and supervised by a researcher well-versed in the method (R.W.), conducted all recalls. To ensure food coding within 95% inter-rater reliability, we randomized the recalls, and a different researcher (T.N.) separately analyzed 10% of the total. As an additional step, we also probed our study sample for unintentional gluten exposure. Our 24 h recalls were manually recorded and then input separately into the Nutrition Data System for Research (NDSR) [52]. We noted that NHANES had two instead of three 24 h recalls, collected several days apart, in previously described methods [27].

#### 2.3.2. Nutrient Assessment

Nutrient values were determined based on whether we collected the data or if they were from NHANES.

In our study sample, we reviewed packaged GF products at the time of the recall for ingredient and nutrient information. Reported consumed items were then analyzed based on foods and ingredients in the NDSR database [52]. If the food available in NDSR did not match the brand of the item consumed, substitutions were made based on ingredients, energy, and macronutrients (±20% margin based on the nutrient label, the industry-allowable variance) [53,54].

NHANES nutrient data were obtained from dietary and food files. Similar to our protocols, to determine nutrients from foods, the NHANES methods specify that the researchers made some substitutes using manufacturers’ labels when similar products were not in their databases. However, differing from our protocol, the NHANES researchers did not use product labels, nor did they additionally probe to determine whether products were GF (which may have altered the reported nutrient results).

Due to the inconsistent nature of consumption among our study sample, we excluded vitamin and minerals taken as supplements from this analysis. We also did not include supplements in the determination of any dietary patterns, nutrient levels, or comparisons with NHANES. Therefore, the nutrient levels were derived from foods, not supplements. The groups’ potential for nutrient deficits or excesses was evaluated based on whether the individual participants, according to demographics, failed to meet U.S.A. estimated average requirements (EARs) or adequate intake (AI) values, and also whether values exceeded the tolerable upper intake level (UL) or fell out of the range of the acceptable macronutrient distribution (AMDR) or estimated energy requirement (EER) [55]. Because the U.S.A. has recently implemented policy to remove added trans fats from the food system and no longer makes specific recommendations in this regard other than to keep trans fats and dietary cholesterol intake as low as possible, we did not include these items in our analysis. However, within the section on potential for excesses, we do note the concerns over the amounts of certain nutrients (sodium, saturated fat, and added sugar as percentages of kcal) that should be limited and are specifically still tracked in the tool suggested by the 2020–2025 Dietary Guidelines for Americans [56] but do not have specific EARs, AIs, ULs, or AMDRs.

### 2.4. Statistical Analysis

Testing for normality and skewness, as well as for homogeneity versus heterogeneity, was performed, with adjustments as applicable. Our study sample was much smaller compared with the NHANES sample, where there could be large differences in variances, and often the expected cell values were small (<5) or even zero. However, it was assumed that the values of the larger population from which all samples were drawn from were normally distributed. Two-sample *t*-tests were used, where applicable, for equal and unequal variances. A variance ratio test was used to determine whether or not the sample variances were equal. Although exact testing can be used for samples of any size, this can be challenging because of the use of resources required. However, these tests were utilized here to calculate the exact probability of observing the statistical differences in categorical variables because of the large difference between the sample sizes. Testing of categorical variables was performed using Fisher’s exact test, expanded from 2 × 2 to 2 × *r* matrices. For the exact test calculations, the weighted percentages of the population were multiplied by the actual population values and rounded to the nearest value. Values for the comparison of nutrients were only shown if at least one sample group’s proportions were 10% or greater. However, all nutrients with recommended requirements were calculated, where approximately 50 variables were hypothesis-tested. A Bonferroni correction was applied using 50 as the denominator for these comparisons. Therefore, the Bonferroni-corrected significance levels went from *p* = 0.05, 0.01, and 0.001 to *p* = 0.001, 0.002, and 0.0002, respectively. All statistical analyses were performed using Stata, version 18 (College Station, TX, U.S.A.).

## 3. Results

### 3.1. Demographics

Compared with the general population in NHANES, our study sample was more likely to be non-Hispanic white, with lower incidence of overweight and obesity. Comparatively, NHANES was significantly more gender-balanced, racially/ethnically diverse, and more likely to be younger, with a higher incidence of overweight and obesity. See Table 1 for the adult population and Table 2 for the teen population. Among study adults, 90% had seen a registered dietitian at least once (16% actively). Among study teens, 70% had seen a registered dietitian at least once (26.7% actively).

### 3.2. Nutrient Status

#### 3.2.1. Nutrients Consumed Below Recommendations

Compared with NHANES, our study participants had fewer nutrients with potential deficits among a significant proportion (10% or more). Still, inclusive of all U.S.A. nutrient recommendations, there were many nutrients of concern for either our study participants or NHANES participants. For our study adults, these included fiber, carbohydrates, linoleic acid, linolenic acid, calcium, iron, magnesium, potassium, and vitamins E, D, C, and K, along with thiamine, choline, and folate (see Table 3). For our study teens, compared to adults, there were more nutrients with potential for deficits, including energy, fiber, carbohydrates, linoleic acid, linolenic acid, calcium, iron, magnesium, phosphorus, potassium, zinc, and vitamins A, E, D, C, B12, and K, along with thiamine, riboflavin, choline, and folate (see Table 4).

Since the estimated average requirement (EAR) indicates the amount needed to maintain health for half of the population using observational or clinical data, not meeting this level suggests a higher probability of overt deficiency occurring. For at least 10% of our study adults, the nutrients that fell into that category of concern included calcium, iron, magnesium, and vitamins E, D, and C, along with thiamin, choline, and folate. For at least 10% of our study teens, the nutrients that fell into that category of concern included calcium, iron, magnesium, phosphorus, zinc, and vitamins A, E, D, C, and B12, along with thiamin, riboflavin, and folate.

For most nutrients, the percent of our study participants falling below the EAR was similar to or lower than that reported by NHANES. However, there were exceptions where a greater proportion of our study participants appeared to be at greater risk, which may be specific to a GFD. For our study adults, folate was the only nutrient of concern, although a higher proportion of the study adults failed to meet the AMDR percentage for carbohydrates as well. For our study teens, only folate was a significant concern.

To determine generalizability, we compared our study adults against NHANES adults who reported CeD and a GFD. Demographically, as compared with the NHANES general population, our study adults were similar to the population that had reported CeD and a GFD (Appendix A, Table A1). In terms of nutrient adequacy, we found that, in most cases, the proportion of NHANES participants below recommendations was similar or greater, with the exception of carbohydrates. However, although fewer nutrients were of concern as compared with NHANES participants who did not have CeD, compared with our study adults, those identified with CeD and a GFD had a slightly higher proportion at risk of deficiency. Based on EAR levels, these included vitamin E, as well as niacin. Notably, there was no significant difference between our study adults and the NHANES participants with CeD and a GFD for the proportion not meeting the folate EAR recommendation. There were not enough NHANES teens in this category for comparison (see Table 5).

#### 3.2.2. Nutrients Consumed Exceeding Recommendations

Compared with NHANES, our study participants had similar numbers of nutrients where a significant proportion (10% or greater) of participants exceeded recommendations. However, overall, there were few nutrients of concern for excess among either our study or the NHANES participants. For our study adults, these included total fat, linolenic acid, and niacin (see Table 6). For our study teens, they included total fat, niacin, vitamin A and, interestingly, folate (see Table 7).

The tolerable upper intake level (UL) indicates the amount where exceeding this level could present a higher probability of harm and be of particular concern, especially when present in a significant proportion of the population (10% or more). For at least 10% of our study adults, niacin (vitamin B3) exceeded this level. For at least 10% of our study teens, niacin (vitamin B3), vitamin A, and folate exceeded this level. Among the nutrients exceeding the UL that appeared to be related to a GFD were niacin for our study adults, as well as total fat exceeding the AMDR percentage, and niacin and vitamin A for our study teens.

To determine generalizability, we compared our study adults against NHANES adults with CeD and adhering to a GFD. There were no significant differences in nutrient consumption above the recommendations between the two groups (see Table 8).

#### 3.2.3. Nutrients of Concern Encouraged to Be Limited

Although there is no explicit upper limit for certain nutrients that are encouraged to be limited, we noted that the consumption of sodium, saturated fat, and added sugar, which are national nutrients of concern, was high for all groups, where the 2020–2025 Dietary Guidelines for Americans suggest that sodium intake be encouraged to be below 2500 mg (or below 1500 mg for certain groups), saturated fat be less than 10% of energy, and added sugars be less than 10% of energy (or lower for optimal health).

Compared with the general population of NHANES, saturated fat consumption as a percent of energy for our study adults was higher (see Table 9 and Table 10). However, across ages and categories, the study sample consumed concerning amounts, but less sodium, and similarly high levels of added sugars (see Table 9, Table 10 and Table 11).

## 4. Discussion

Overall, this study suggests that individuals with CeD adhering to a GFD, particularly those who are affiliated with a CeD-specific center, have a higher nutritional quality in their diet than the typical American. These are new and perhaps unsurprising findings. This result may be due to the fact that CeD treatment requires attention to the diet and may encourage greater awareness of dietary quality and health consciousness. The majority of those in the study sample had an affiliation with the center and had seen a registered dietitian at least once. Those who have an affiliation with a CeD-specific center may be even further engaged with the nutritional intake of their diet than those in the general population; often, those attending such centers are greatly motivated to improve their outcomes. It is a longstanding practice for newly diagnosed CeD patients to be referred to a registered dietitian nutritionist—although, despite recommendations, many patients may not continue regular visits [57,58,59,60]. Although regular dietitian follow-up is the standard of care and encouraged, those with CeD maintaining a GFD could still make changes to improve their nutrient intake and reduce their potential for nutrient deficiencies and excesses. There were numerous nutrients of concern for our study participants, which paralleled issues with the NHANES sample. Troublingly, a significant percentage (>10%) of our study participants may have had potential deficits or excesses of multiple nutrients, especially among teens. Still, only a few nutrients overall were suggestive of overt clinical consequences as compared with NHANES. Most of the nutrients of concern had been noted in prior studies in other countries.

One of our most concerning findings was the potential for overt nutrient deficiency in our study participants. This may be due, in part, to the formulation of GF products, since a large proportion fell below the EAR for folate. Folate has been and continues to potentially be problematic for individuals in the U.S.A. adhering to a GFD, due to its exclusion from the Food Code and the consequent lack of fortification requirements among GF products. Folate is a nutrient that is found naturally in green leafy vegetables, nuts, and legumes. However, the chief food sources of folic acid for those not adhering to a GFD are widely consumed staple products, such as (fortified or enriched wheat-based) grain products. Per the Food Code, these products are commonly enriched with folic acid to accommodate population-wide deficits [61] in folate due to the low consumption of folate in other natural food sources. Consistent with prior recommendations [62], practitioners should assess serum folate levels and consider supplementation, as warranted, particularly for women of childbearing age to prevent neural tube defects. However, folic acid supplementation may have slight health risks for men [63,64,65,66,67,68,69]. Numerous nutrients were of concern because large proportions of our sample, especially the teens, failed to meet the recommendations. Iron is one of those nutrients that are also included in the fortification/enrichment process in staple products, falling especially short for study teens. Carbohydrate intake was also lower for our study participants, with a large percentage not meeting the AMDR. Adequate consumption of carbohydrates may be less immediately pressing than overt folate or iron deficiency. However, lack of carbohydrates may cause individuals to feel fatigue, as carbohydrates are a primary source of energy for the body, and especially for the brain. Carbohydrates also contribute to obtaining an adequate amount of fiber, which is also needed for numerous bodily functions, such as cholesterol clearance and gut transit, among other functions [70]. Identifying these nutrient intake shortfalls is critical for everyone, especially for women of childbearing age and growing teens. While not normative, some manufacturers are choosing to fortify GF breads and other GF products. However, formulations may make those products high in ultra-processed ingredients, which may introduce other health concerns [13,14,15,16,17,18].

Another concern for our study participants relative to NHANES was potentially excessive total fat (% energy) (study adults), as well as niacin and vitamin A (study teens) consumption. Excessive total fat, and especially saturated fat, is associated with the most risk of cardiovascular and other chronic diseases, including nonalcoholic fatty liver disease (NAFLD), which has been linked to individuals with CeD [71]. Where possible, this also includes limiting the consumption of plant-based oils that are high in saturated fat, like coconut and palm oil. The latter are increasingly being added to foods as emulsifiers. Excessive niacin, ironically enough, can mimic symptoms associated with gluten exposure, including nausea and gastrointestinal symptoms [72]. These effects may be seen when individuals are consuming excessive synthetic supplementation, or possibly through foods, perhaps when niacin is added to products like cereals and other packaged foods. Vitamin A, a fat-soluble vitamin, can be toxic when taken in excessive amounts. Toxicity can manifest as fatigue and intestinal distress, mimicking gluten exposure for those with CeD, and even potentially causing birth defects [73]. It is associated with bone structure deformities, including osteoporosis [73,74]. Vitamin A may also be added to packaged foods. Therefore, practitioners should strongly encourage patients to read labels in order to understand sources of nutrients, so as to limit some and to encourage others, but more generally to consume an overall healthy, minimally processed dietary pattern. Although there were no ULs developed, both our study sample and NHANES showed concerns for excessive consumption of saturated fat, sodium, and added sugar. Reductions of all three would be warranted. In particular, our study sample consumed significantly more saturated fat (study adults) than the NHANES participants, which suggests that concerns may be focused there for the GFD as well.

Our study sample had numerous nutrient concerns, which often overlapped with those noted in the NHANES sample. These included potential shortfalls of vitamin D and E, along with other fat-soluble vitamins, which can impact numerous metabolic functions, including bone density, skin, and immunity. These concerns likely reflect American dietary trends of low overall consumption of fish, plant oils, whole grains, legumes, seeds, nuts, and vegetables more broadly, especially leafy greens [57]. Nutrient shortfalls like these do not necessarily result in overt and immediate manifestation, but they may contribute to a lack of wellbeing and optimal outcomes over the long term.

Overall, compared to NHANES, our study participants had a better nutritional profile. This aligns with observations comparing diet quality [13]. Despite having many more nutrients of concern than adults, our study teens had a better nutritional profile than participants in NHANES as well. These findings reiterate the need to appreciate overall societal nutrient concerns, the limits of GF products, and lack of fortification, as well as individual eating patterns. Recognizing that inferior nutrient consumption may be endemic for the general population is important, due to the potential for suboptimal absorption of nutrients even after adhering to a GFD for individuals with CeD. It is also important to recognize additional potential shortfalls in the GFD due to the current policy on fortification in the U.S.A. Importantly, the population at the CeD center in this sample, based on the differences in meeting the suggested nutrient requirements, are eating more healthily in general, yet they are at risk of inadequate consumption of many key nutrients—like many in the population. This study suggests that more awareness may be needed to ensure nutritional balance for those who have CeD and maintain a GFD.

Food and drink were the main subjects of the nutrient analysis, not supplements. This decision was made primarily due to the inconsistent nature of our study participants’ consumption of vitamins and other supplements. However, it is important to recognize that potential nutrient deficits in those maintaining a GFD may not be simply addressed by consumption of a multivitamin mineral supplement, due to formulations that may be inadequate for certain nutrients and beyond recommendations for others. Continuing to assess individuals based on their individual needs is critical. Nutrients obtained through minimally processed foods generally mitigate the risk of toxic side effects and should be encouraged. Therefore, all practitioners specializing in CeD should be trained and encouraged to promote nutritional adequacy by improving self-efficacy in eating a high-quality, minimally processed diet.

### Strengths and Limitations

A strength of this research was its concordance with the findings of other studies, enabling us to compare it to a representative sample, including a subset that, though imperfect, reported having CeD and maintaining a GFD. Prior studies have suggested that individuals with CeD adhering to a GFD may be deficient in carbohydrates (% energy), calcium, iron, magnesium, selenium, zinc, polyunsaturated fat, and B vitamins, including folate, riboflavin, and B12 [2,15,19,20,21,22]. Here, we note that most of these potential deficiencies, with the exception of vitamin A, folate, and carbohydrates, were generally similar or had lower prevalence among those adhering to a GFD as compared with NHANES. The same studies, as previously referenced, also reported high intake of total fat and saturated fats as a percentage of total energy among individuals with CeD and maintaining a GFD, which we also observed [2,15,19,20,21,22]. Notably, though, these studies were conducted in countries other than the U.S.A., suggesting global concerns with the GFD and with general nutrient intake. Therefore, this adds to the literature by suggesting that, despite the U.S.A.’s efforts to safeguard the food system and prevent overt nutrient deficiencies, nutrients are lacking for individuals with CeD who require a GFD.

This study reaffirms the importance of ongoing referral to dietitians trained in maintaining a GFD for nutritional education as a sound recommendation due to adherence concerns as well as nutrient concerns. Although the GFD is the prescription for the treatment of CeD, mastering adherence to the GFD requires nutritional education, which takes time and repeated visits. However, many people only see a dietitian once, if at all, despite recommendations to go regularly [57,58,59,60].

We were able to conduct a detailed probing of gluten-free products, including capturing nutrient label information and replicating the profiles of foods consumed. This may have been a greater level of detail than was captured with the NHANES participants.

We were also able to recruit 50 adults and 30 teens adhering to a GFD who attested to biopsy-confirmed diagnoses of CeD. Although smaller than other studies, within the U.S.A. this is a moderate sized cohort for a study where dietary recalls are completed with individuals with CeD. Notably, even though NHANES represents a nationally representative sample and had nearly 30,000 participants, representative of the entire non-institutionalized population of the U.S.A (~300 million), only 25 adults and only 1 teen met the inclusion criteria and were included in that sampling. Our recruited sample size reflects the need for future studies of larger cohorts, with diverse characteristics, to determine whether other factors may mitigate the findings.

Relatedly, there were some limitations in undertaking this research. Our study sample may not necessarily be representative of the larger CeD population. This made it difficult to conduct subgroup analysis or conduct analyses where additional confounders were controlled. Moreover, the size difference between our study sample and the NHANES participants without CeD is significant and will often result in variance size differences requiring specialized statistics. However, we did find differences aligning with our hypotheses of potential nutrient shortfalls for our study sample, where the rest of the population had more opportunity for exposure to gluten-containing foods. Moreover, our study participants generally had higher adherence to nutritional recommendations overall, which suggests that differences showing potential deficits are, indeed, especially concerning. As a sensitivity analysis, our comparison against NHANES participants reporting CeD and a GFD showed very few nutrient differences, suggesting higher reliability with our results. The overall use of a conservative Fisher’s test with a Bonferroni correction requires a greater level of difference in order to show significance. Therefore, the significant differences found here suggest that these results may shed some light on what is happening within this population. Still, the NHANES data are designed to be representative of the U.S.A. population based on age, gender, income levels, and race/ethnicity, not of medical conditions like CeD, without oversampling. In addition, there were fewer differences between our study sample and those in NHANES reporting CeD and a GFD, who were much fewer than the larger general NHANES group, which suggests that our results may offer a degree of wider generalizability to those who have a diagnosis of CeD and maintain a GFD. Further study to obtain a nationally representative sample of individuals with CeD may be warranted.

Another potential limitation was that the NHANES protocols were not designed to specifically for a GFD, which may limit the comparison of nutrients from those maintaining a GFD. The NHANES participants were asked whether they adhered to a GFD or not after undergoing a 24 h recall, not prior to the recall [26,27]. This meant that values for some items, like pizza or pasta, may have been recorded as regular, gluten-containing foods, whose nutritional profiles differ from those used in a GFD (e.g., assumed folate fortification of grains). In addition, the researchers who provided nutritional information on GF products in NHANES at the time had a limited nutritional database of GF products. Cumulatively, this may have resulted in inaccurate NHANES nutrient values for those adhering to a GFD.

We also acknowledge that a small number of participants were included in the general NHANES category who maintained a GFD without a CeD diagnosis. This group may have potentially had nutritional similarities to our study sample, reducing the significance of differences found between the results. However, due to the relatively small number of this group compared with the total number of NHANES participants, it would be unlikely to change the overall noted differences.

The 24 h recall is a validated tool for recording diet intake. However, there are inherent limitations with its use, such as over- and underreporting. These limitations are somewhat mitigated by using aggregate values and multiple days of recalls on weekends and weekdays.

Our study sample was mainly female, non-Hispanic white, living in higher-income neighborhoods, and/or highly educated, which may have meant lower availability of UPF, resulting in higher dietary nutrient density. This was also similar to the sample identified within the NHANES database who reported CeD and a GFD. Individuals who are not non-Hispanic white may have different dietary patterns when they adhere to a GFD. There may have also been factors specific to our study sample that drove the observed differences, such as access to expert dietitian care.

## 5. Conclusions

Although nutrient pattern improvements could be made, our study sample had favorable overall nutrient quality when compared with the NHANES sample. Overall, our study sample’s nutrient concerns were potentially extensive and reflective of the larger representative population, with the possible exception of low folate and carbohydrate (sample adults) consumption, with excessive niacin, vitamin A (sample teens), and saturated and total fat consumption (sample adults).

## Figures and Tables

**Table 1 nutrients-16-03067-t001:** Demographics of study adults with CeD and NHANES participants (aged 18 years+) ^1^.

	Study Adults *n* = 50	NHANES Adults *n* = 15,777
Gender *n* (%)	***	
Female	42 (84.0)	8009 (50.9)
Age in Years		
Mean (SD)	50.7 (17.8)	46.5 (0.4)
Race/Ethnicity *n* (%)	***	
Non-Hispanic White	47 (94.0)	6818 (66.7)
All Other Races/Ethnicities	3 (6.0)	8957 (33.3)
Body Mass Index Status *n* (%)	***	
Underweight	2 (4.0)	285 (1.7)
Normal	35 (70.0)	4612 (30.2)
Overweight	10 (20.0)	4989 (32.3)
Obesity	3 (6.0)	5730 (35.8)
Education *n* (%)	***	
<High School	4 (8.0)	3554 (16.4)
HS Graduate/Some College	12 (24.0)	7794 (53.9)
College Graduate or Higher	34 (68.0)	3566 (29.7)

^1^ NHANES = National Health and Nutrition Examination Survey (multi-staged, survey-weighted values based on representative sampling). CeD = celiac disease. SD: standard deviation. Two-sample *t*-test with unequal variances was used for age differences. Fisher’s exact test was used for gender, race/ethnicity, BMI, and education differences. *** *p* < 0.001.

**Table 2 nutrients-16-03067-t002:** Demographics of study teens with CeD and NHANES participants (aged 13–17 years) ^1^.

	Study Teens *n* = 30	NHANES Teens *n* = 2296
Gender *n* (%)	**	
Female	24 (80.0)	1115 (51.7)
Age in Years	*	
Mean (SD)	15.7 (1.5)	15.0 (0.1)
Race/Ethnicity *n* (%)	***	
Non-Hispanic White	29 (96.7)	623 (56.4)
All Other Races/Ethnicities	1 (3.3)	1673 (43.6)
Body Mass Index Status *n* (%)	***	
Underweight	2 (6.7)	58 (2.2)
Normal	28 (93.3)	1390 (63.3)
Overweight	0 (0.0)	369 (14.5)
Obesity	0 (0.0)	460 (20.0)
Education *n* (%)	*	
<Middle School	0 (0.0)	13 (0.6)
Middle School (6th–8th Grades)	7 (23.3)	1054 (44.8)
High School (9th–12th Grades)	22 (73.3)	1204 (53.7)
Some College	1 (3.3)	25 (1.0)

^1^ NHANES = National Health and Nutrition Examination Survey (multi-staged, survey-weighted values based on representative sampling). CeD = celiac disease. SD: standard deviation. Two-sample *t*-test with unequal variances was used for age differences. Fisher’s exact test was used for gender, race/ethnicity, BMI, and education differences. * *p* < 0.05; ** *p* < 0.01; *** *p* < 0.001.

**Table 3 nutrients-16-03067-t003:** Number (%) of study adults with CeD and weighted percentage and number of NHANES participants (aged 18+ years) with dietary nutrient consumption below recommendations for at least 10% of either sample ^1^.

	Study Adults *n* = 50 *n* (%)	NHANES Adults *n* = 15,616 *n* (Weighted %)	
Macronutrients, Fiber, and Fatty Acids			
Energy < TEE	0 (0.0%)	5247.1 (33.6%)	***
Fiber < AI	30 (60.0%)	13,959.6 (88.5%)	***
Protein < EAR	1 (2.0%)	3069.6 (19.6%)	***
**Carbohydrate < %AMDR**	**36 (72.0%)**	**5199.1 (33.0%)**	***
Linoleic Acid < AI	12 (24.0%)	6934.5 (44.0%)	
Linoleic Acid < %AMDR	10 (20.0%)	3522.0 (22.3%)	
Linolenic Acid < AI	13 (26.0%)	6688.1 (42.4%)	
Linolenic Acid < %AMDR	16 (32.0%)	6334.5 (40.2%)	
Minerals and Vitamins			
Calcium < EAR	22 (44.0%)	7570.9 (48.0%)	
Iron < EAR	5 (10.0%)	1053.9 (6.7%)	
Magnesium < EAR	7 (14.0%)	8999.6 (57.0%)	***
Potassium < AI	17 (34.0%)	10,435.1 (66.1%)	***
Zinc < EAR	4 (8.0%)	4238.7 (28.9%)	***
Vitamin A < EAR	3 (6.0%)	8096.7 (51.3%)	***
Vitamin E < EAR	26 (52.0%)	15,550.2 (98.6%)	***
Vitamin D < EAR	37 (74.0%)	14,205.2 (90.0%)	*
Vitamin C < EAR	12 (24.0%)	8292.9 (52.6%)	***
Thiamin < EAR	11 (22.0%)	2246.8 (14.2%)	
Niacin < EAR	0 (0.0%)	2168.1 (13.7%)	*
Vitamin B6 < EAR	2 (4.0%)	2836.1 (18.0%)	
Vitamin B12 < EAR	3 (6.0%)	2213.9 (14.0%)	
Choline < AI	38 (76.0%)	13,689.9 (86.8%)	
Vitamin K < AI	12 (24.0%)	10,274.3 (65.1%)	***
**Folate < EAR**	**23 (46.0%)**	**3466.3 (22.0%)**	*******

^1^ Bold indicates that our study participants, as compared with NHANES participants, had proportionately more individuals with dietary nutrient consumption below recommendations. Nutrients are only shown if the percentage below recommendations for either our study or NHANES was 10% or greater. Other nutrients were examined and did not meet this threshold. Notes: CeD = celiac disease. NHANES = National Health and Nutrition Examination Survey (multi-staged, survey-weighted values based on representative sampling). EAR = estimated average requirement. AI = adequate intake. AMDR = acceptable macronutrient distribution range. TEE = total estimated energy. Bonferroni correction was applied. Fisher’s exact test values were used for statistical significance; * *p* < 0.001, *** *p* < 0.00002.

**Table 4 nutrients-16-03067-t004:** Number (%) of study teens with CeD and weighted number of NHANES participants (aged 13–17 years) with dietary nutrient consumption below recommendations for at least 10% of either sample ^1^.

	Study Teens *n* = 30 *n* (%)	NHANES Teens *n* = 2296 *n* (Weighted %)	
Macronutrients, Fiber, and Fatty Acids			
Energy < TEE	12 (40.0%)	1361.3 (59.8%)	
Fiber < AI	24 (80.0%)	2251.9 (98.1%)	***
Protein < EAR	0 (0.0%)	449.1 (19.7%)	
Carbohydrate < %AMDR	11 (36.7%)	359.7 (15.7%)	
Linoleic Acid < AI	9 (30.0%)	1057.3 (46.0%)	
Linoleic Acid < %AMDR	5 (16.7%)	627.2 (27.3%)	
Linolenic Acid < AI	6 (20.0%)	1173.5 (51.1%)	*
Linolenic Acid < %AMDR	12 (40.0%)	1204.4 (52.5%)	
Minerals and Vitamins			
Calcium < EAR	17 (56.7%)	1413.9 (61.6%)	
Iron < EAR	9 (30.0%)	265.3 (11.6%)	
Magnesium < EAR	14 (46.7%)	1668.8 (72.7%)	
Phosphorus < EAR	9 (30.0%)	713.3 (31.1%)	
Potassium < AI	14 (46.7%)	1558.6 (67.9%)	
Zinc < EAR	3 (10.0%)	676.2 (29.4%)	*
Vitamin A < EAR	5 (16.7%)	1210.5 (52.7%)	***
Vitamin E < EAR	18 (60.0%)	2290.4 (99.8%)	***
Vitamin D < EAR	27 (90.0%)	2025.0 (88.2%)	***
Vitamin C < EAR	8 (26.7%)	1220.7 (53.2%)	
Thiamin < EAR	5 (16.7%)	310.5 (13.5%)	
Riboflavin < EAR	3 (10.0%)	203.6 (8.9%)	
Niacin < EAR	0 (0.0%)	412.9 (18.0%)	
Vitamin B6 < EAR	2 (6.7%)	360.1 (15.7%)	
Vitamin B12 < EAR	4 (13.3%)	282.6 (12.3%)	
Choline < AI	25 (83.3%)	2086.1 (90.9%)	
Vitamin K < AI	12 (40.0%)	1637.1 (71.3%)	***
**Folate < EAR**	**15 (50.0%)**	**514.1 (22.4%)**	*

^1^ Bold indicates that our study participants had proportionately more individuals with dietary nutrient consumption below recommendations as compared with at least one NHANES group. Nutrients are only shown if the percentage below recommendations for either our study or NHANES was 10% or greater. Other nutrients were examined and did not meet this threshold. Notes: CeD = celiac disease. NHANES = National Health and Nutrition Examination Survey (multi-staged, survey-weighted values based on representative sampling). EAR = estimated average requirement. AI = adequate intake. AMDR = acceptable macronutrient distribution range. TEE = total estimated energy. Bonferroni was correction applied. Fisher’s exact test values were used for statistical significance; * *p* < 0.001, *** *p* < 0.00002.

**Table 5 nutrients-16-03067-t005:** Number (%) of study sample adults with CeD and weighted number of NHANES participants (aged 18+ years) reporting CeD and a GFD with dietary nutrient consumption below recommendations for at least 10% of either sample ^1^.

	Study Adults *n* = 50 *n* (%)	NHANES CeD & GFD *n* = 25 *n* (Weighted %)	
Macronutrients, Fiber, and Fatty Acids			
Protein < EAR	1 (2.0%)	4.2 (16.6%)	
**Carbohydrate < %AMDR**	**36 (72.0%)**	**8.1 (32.6%)**	*
Fiber < AI	30 (60.0%)	20.1 (80.4%)	
Linoleic Acid < AI	12 (24.0%)	11.1 (44.4%)	
Linoleic Acid < %AMDR	10 (20.0%)	4.1 (16.5%)	
Linolenic Acid < AI	13 (26.0%)	10.1 (40.3%)	
Linolenic Acid < %AMDR	16 (32.0%)	10.7 (42.7%)	
Minerals and Vitamins			
Calcium < EAR	22 (44.0%)	12.2 (48.8%)	
Iron < EAR	5 (10.0%)	5.0 (20.1%)	
Magnesium < EAR	7 (14.0%)	9.8 (39.3%)	
Potassium < AI	17 (34.0%)	14.1 (56.4%)	
Vitamin A < EAR	3 (6.0%)	5.0 (20.0%)	
Vitamin E < EAR	26 (52.0%)	23.2 (92.6%)	*
Vitamin D < EAR	37 (74.0%)	23.1 (92.4%)	
Vitamin C < EAR	12 (24.0%)	8.5 (34.1%)	
Thiamin < EAR	11 (22.0%)	4.2 (17.0%)	
Niacin < EAR	0 (0.0%)	6.3 (25.1%)	*
Vitamin B6 < EAR	2 (4.0%)	4.8 (19.2%)	
Vitamin B12 < EAR	3 (6.0%)	3.2 (12.8%)	
Choline < AI	38 (76.0%)	20.2 (80.9%)	
Vitamin K < AI	12 (24.0%)	9.2 (36.7%)	
Folate < EAR	23 (46.0%)	4.1 (16.3%)	

^1^ Bold indicates that our study participants, as compared with NHANES participants, had proportionately more individuals with dietary nutrient consumption below recommendations. Nutrients are only shown if the percentage below recommendations for either our study or NHANES was 10% or greater. Other nutrients were examined and did not meet this threshold. Notes: NHANES = National Health and Nutrition Examination Survey (multi-staged, survey-weighted values based on representative sampling). CeD = celiac disease. EAR = estimated average requirement. AI = adequate intake. AMDR = acceptable macronutrient distribution range. Fisher’s exact test values were used for statistical significance; * *p* < 0.001.

**Table 6 nutrients-16-03067-t006:** Number (%) of study adults with CeD and weighted number of NHANES participants (aged 18+ years) with dietary nutrient consumption above recommendations for at least 10% of either sample ^1^.

Nutrient Consumption Above Recommendations	Study Adults *n* = 50 *n* (%)	NHANES Adults *n* = 15,616 *n* (Weighted %)	
**Total Fat > %AMDR**	**40 (80.0%)**	**6726.3 (42.6%)**	*******
Linoleic Acid > %AMDR	4 (8.0%)	1731.4 (11.0%)	
Linolenic Acid > %AMDR	5 (10.0%)	1201.6 (7.6%)	
**Niacin (Vitamin B3) > UL**	**28 (56.0%)**	**2842.2 (18.0%)**	*******

^1^ Bold indicates that our study had proportionately more individuals with dietary nutrient consumption above the recommended upper levels as compared with NHANES. Nutrients are only shown if the percentage exceeding recommendations for either our study sample or NHANES was 10% or greater. Other nutrients were examined and did not meet this threshold. Notes: NHANES = National Health and Nutrition Examination Survey (survey-weighted values). CeD = celiac disease. AMDR = acceptable macronutrient distribution range. UL = tolerable upper intake level. Fisher’s exact test values were used for statistical significance; *** *p* < 0.00002.

**Table 7 nutrients-16-03067-t007:** Number (%) of study teens with CeD and weighted number of NHANES participants (aged 13–17 years) with dietary nutrient consumption above recommendations for at least 10% of either sample ^1^.

Nutrient Consumption Above Recommendations	Study Teens *n* = 30 *n* (%)	NHANES Teens *n* = 2296 *n* (Weighted %)	
Total Fat > %AMDR	20 (66.7%)	862.9 (37.6%)	
**Niacin > UL**	**24 (80.0%)**	**720.9 (31.4%)**	*******
**Vitamin A > UL**	**3 (10.0%)**	**2.0 (0.1%)**	*******
Folate > UL	6 (20.0%)	470.0 (20.5%)	

^1^ Bold indicates that our study participants had proportionately more individuals with dietary nutrient consumption above the recommended upper levels as compared with the NHANES participants. Nutrients are only shown if the percentage exceeding recommendations for either our study or NHANES was 10% or greater. Other nutrients were examined and did not meet this threshold. Notes: CeD = celiac disease. NHANES = National Health and Nutrition Examination Survey (multi-staged, survey-weighted values based on representative sampling). AMDR = acceptable macronutrient distribution range. UL = tolerable upper intake level. Bonferroni correction was applied. Fisher’s exact test values were used for statistical significance; *** *p* < 0.00002.

**Table 8 nutrients-16-03067-t008:** Number (%) of study adults with CeD and weighted number of NHANES participants (aged 18+ years) who had CeD and a GFD with dietary nutrient consumption above recommendations for at least 10% of either sample ^1^.

Nutrient Consumption Above Recommendations	Study Adults *n* = 50 *n* (%)	NHANES CeD & GFD *n* = 25 *n* (Weighted %)
Total Fat > %AMDR	40 (80.0%)	11.0 (43.9%)
Linoleic Acid > %AMDR	4 (8.0%)	5.2 (20.9%)
Linolenic Acid > %AMDR	5 (10.0%)	2.0 (8.0%)
Niacin (Vitamin B3) > UL	28 (56.0%)	4.6 (18.5%)
Folate > UL	0 (0.0%)	2.5 (10.0%)

^1^ Nutrients are only shown if the percentage exceeding recommendations for either our study sample or NHANES was 10% or greater. Other nutrients were examined and did not meet this threshold. Notes: NHANES = National Health and Nutrition Examination Survey (multi-staged, survey-weighted values based on representative sampling). CeD = celiac disease. AMDR = acceptable macronutrient distribution range. UL = tolerable upper intake level. Bonferroni correction was applied. Fisher’s exact test values were used for statistical significance; No differences were found to be significantly different in this table.

**Table 9 nutrients-16-03067-t009:** Number (%) of study adults with CeD and weighted number of NHANES participants (aged 18+ years), and amounts of nutrients of concern ^1^.

Nutrients to Limit	Study Adults *n* = 50	NHANES Adults *n* = 15,777	
Sodium ^2^ [Mean mg (SD)]	3359 (940)	6667 (2915)	***
**Saturated Fat [Mean % Energy (SD)]**	**12.8 (3.1)**	**10.9 (3.2)**	******
Added Sugar [Mean % Energy (SD)]	9.5 (5.4)	12.7 (8.4)	**

^1^ Nutrients of concern are those where there are no set maximums, but are meant to be limited, where bold indicates that our study had proportionately more individuals with dietary nutrient consumption as compared with NHANES. ^2^ Variances unequal. Notes: CeD = celiac disease. NHANES = National Health and Nutrition Examination Survey (multi-staged, survey-weighted values based on representative sampling). Two-tailed *t*-test with equal or, if designated, unequal variances was used to determine statistical significance. Bonferroni correction was applied; ** *p* < 0.0002, *** *p* < 0.00002.

**Table 10 nutrients-16-03067-t010:** Number (%) of study adults with CeD and weighted number of NHANES participants (aged 18+ years) who had CeD and a GFD, and amounts of nutrients of concern ^1^.

Nutrients to Limit	Study Adults *n* = 50	NHANES CeD and GFD *n* = 25	
Sodium ^2^ [Mean mg (SD)]	3359 (940)	5805 (2522)	**
**Saturated Fat [Mean % Energy (SD)]**	**12.8 (3.1)**	**10.1 (2.7)**	*****
Added Sugar [Mean % Energy (SD)]	9.5 (5.4)	9.6 (4.4)	

^1^ Nutrients of concern are those where there are no set maximums, but are meant to be limited, where bold indicates that our study had proportionately more individuals with dietary nutrient consumption as compared with NHANES. ^2^ Variances unequal. Notes: NHANES = National Health and Nutrition Examination Survey (multi-staged, survey-weighted values based on representative sampling). CeD = celiac disease. Two-tailed *t*-test with equal or, if designated, unequal variances was used to determine statistical significance. Bonferroni correction was applied; * *p* < 0.001, ** *p* < 0.0002.

**Table 11 nutrients-16-03067-t011:** Number (%) of study teens with CeD and weighted number of NHANES Participants (aged 13–17 years), and amounts of nutrients of concern ^1^.

Nutrient Consumption	Study Teens *n* = 30	NHANES Teens *n* = 2296	
Sodium ^2^ [Mean mg (SD)]	3394 (1220)	6393 (2837)	***
Saturated Fat ^2^ [Mean % Energy (SD)]	13.6 (4.2)	11.2 (2.9)	
Added Sugar [Mean % Energy (SD)]	11.2 (6.8)	15.6 (7.5)	

^1^ Nutrients of concern are those where there are no set maximums, but are meant to be limited. ^2^ Variances unequal. Notes: CeD = celiac disease. NHANES = National Health and Nutrition Examination Survey (multi-staged, survey-weighted values based on representative sampling). Two-tailed *t*-test with equal or, if designated, unequal variances was used to determine statistical significance. Bonferroni correction was applied; *** *p* < 0.00002.

## Data Availability

The data presented in this study may be available upon reasonable request from the corresponding author. The data are not publicly available due to confidentiality concerns.

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
