# Peer review of "Dietary Adherence to Recommendations among a Cohort of Adults and Teens with Celiac Disease Maintaining a Gluten-Free Diet Compared to a Nationally Representative Sample: A Cross-Sectional Study"

_nutrients, 2024, doi:10.3390/nu16183067_

Round 1

Reviewer 1 Report

Comments and Suggestions for Authors

The manuscript by Cadenhead et al. aims to explore the nutritional profile of individuals with biopsy confirmed celiac disease who were on a gluten-free diet, compared with a control group from a large American survey. The authors conclude that, compared with the average American, celiac subjects had a similar or better profile from a qualitative point of view, a result not unexpected given the involvement of registered dietitians.

The sampling methodology and dietary assessment have been carefully described, and the conclusions are supported by the results obtained. Overall, the manuscript is well written, and references are appropriate. The figures are of acceptable quality, but perhaps with too small characters.

My only serious concern is related to the statistical procedures and data presentation which in my opinion require some clarification.

Page 3, lines 115-116. Was the decision to recruit only 50 adults and 30 teens dictated by some calculation of sample size or power analysis? There is a remarkable imbalance between the number of cases and controls.

Page 4, line 198 and page 5, line 199. Why was a parametric test (Student’s t-test) chosen to compare means despite the limited size of the Study Sample? Most of significant differences could therefore be attributed to the disproportionately larger size of the NHANES control group (except for Tables 5 and 8).

Page 5. Given the size disproportion between the Study Sample and the NHANES group, many of the statistically significant differences are probably due to such imbalanced data. For example, in Table 1 the age difference is statistically significant because there were over 15,000 controls. If the number of controls had been equal to the number of cases, the p-value would have been only 0.098. The same is true for Fiber at page 7, Table 4, and in many other cases in the following tables.

Moreover, some of the cells in the contingency tables contain zeros. In this case the independence assumption is violated, and it may happen that neither the chi-square test nor Fisher's exact test are appropriate to calculate the p-value. For instance, on page 6, Table 2, the participants with less than Middle School were zero in the Study Sample and 13 in the NHANES Group. The p-value is equal to 1.0 and should not be marked as highly significant (***) as reported in the Table. However, I acknowledge that for iron and folate in Table 4 any alternative choice in the statistical procedure would not have changed the results.

Finally, some percentages in parentheses, in the column of NHANES controls, seem quite different from those that can be calculated directly. Is there any weighting involved?

I think it is advisable to request a statistician to review the manuscript.

Minor remarks

The nutrients considered in the study did not include cholesterol, which in the past had raised some concerns about gluten-free diets.

Although I am not qualified in English language, the text seems excellent to me. I have some doubts about page 2, line 78, where the expression “to council patients” is used instead of “counsel”, but perhaps this is due to my ignorance.

Reviewer 2 Report

Comments and Suggestions for Authors

Overall, the paper is quite well written and the study is correctly designed. However I have some hesitations regarding methodology and statistical parts.

1. Please explain which method exactly was used to calculate differences e.g. in distribution of education level and population: Study vs NHANES? In statistical methods I found that Fisher’s exact tests were performed. However –one p-value should be given while in table 2 separate p-values were calculated for each educational level. So that frequency of occurring “HS Graduate/ Some College” and “College Graduate or higher” categories was found to differ significantly between two population. Please explain which tests were used after performing omnibus Fisher exact tests to find pairwise differences? Were some corrections for multiple comparisons implemented? Similarly in Table 2 – please get a look at significant differences for education, Body Mass Index Status n (%) …

2.       Some variables are separated together in tables in non-intuitive manner. For example, why in Tables 1 and 2 sex and age are grouped together to form one block. Please add the information in “Age in years (SD)” which statistic was calculated as a measure of central tendency? Was it a mean value? Please insert such information in the table, e.g. “Age in years Mean (SD)”

3.       Style should be improved in some places, e.g. in the sentence: “Overall, compared to NHANES our Study participants had a better nutritional profile than NHANES” – there is no need to repeat twice “compared to NHANES… than NHANES”

4.       I appreciate that the limitation part is very extensive and well described, however the strengths are not reported explicitly in discussion. I suggest to put more emphasize in the discussion also for the strengths of the study.

5.       In the introduction, the authors tried to justify the use of the NHANES population as a reference population suitable for comparison, however I did not find any information about NHANES in the title of the article. I suggest adding  in the title information about the NHANES population as a reference for the comparisons made, and also better justifying/emphasizing the meaning of such analyses based on the comparison with NHANES in the discussion as well.

6.       Taking into account some demographic and socio-economic differences between Study and NHANES populations, did you consider to perform more advanced statistical analysis besides univariable? Other covariates and modifiers can affect the difference between nutrient consumption and should be incorporated to the models. Please address this problem in the results and discussion.

7.       Statistical methods should be improved and described in detail.

Round 2

Reviewer 2 Report

Comments and Suggestions for Authors

I accept the Authors' explanation that "the most realistic calculations we could make was to compare the samples by age in this article...". Authors addressed all my remaining comments. I accept the publication in a current form